# Challenge Accepted? Individual Performance Gains for Motor Imagery Practice with Humanoid Robotic EEG Neurofeedback

**DOI:** 10.3390/s20061620

**Published:** 2020-03-14

**Authors:** Mareike Daeglau, Frank Wallhoff, Stefan Debener, Ignatius Sapto Condro, Cornelia Kranczioch, Catharina Zich

**Affiliations:** 1Neurocognition and Functional Neurorehabilitation Group, Neuropsychology Lab, Department of Psychology, School of Medicine and Health Sciences, Carl von Ossietzky University Oldenburg, 26111 Oldenburg, Germany; cornelia.kranczioch@uol.de (C.K.); catharina.zich@uni-oldenburg.de (C.Z.); 2Institute for Assistive Technologies, Jade University of Applied Science, 26389 Oldenburg, Germany; frank.wallhoff@jade-hs.de (F.W.); saptocondro@gmail.com (I.S.C.); 3Neuropsychology Lab, Department of Psychology, School of Medicine and Health Sciences, Carl von Ossietzky University Oldenburg, 26111 Oldenburg, Germany; stefan.debener@uol.de; 4Cluster of Excellence Hearing4All, Carl von Ossietzky University Oldenburg, 26111 Oldenburg, Germany; 5Research Center Neurosensory Science, Carl von Ossietzky University Oldenburg, 26111 Oldenburg, Germany; 6Department of Clinical and Movement Neuroscience, UCL Queen Square Institute of Neurology, London WC1N 3BG, UK; catharina.zich@uol.de

**Keywords:** BCI, mobile EEG, neurofeedback, robot, motor imagery, ERD/S, individual differences

## Abstract

Optimizing neurofeedback (NF) and brain–computer interface (BCI) implementations constitutes a challenge across many fields and has so far been addressed by, among others, advancing signal processing methods or predicting the user’s control ability from neurophysiological or psychological measures. In comparison, how context factors influence NF/BCI performance is largely unexplored. We here investigate whether a competitive multi-user condition leads to better NF/BCI performance than a single-user condition. We implemented a foot motor imagery (MI) NF with mobile electroencephalography (EEG). Twenty-five healthy, young participants steered a humanoid robot in a single-user condition and in a competitive multi-user race condition using a second humanoid robot and a pseudo competitor. NF was based on 8–30 Hz relative event-related desynchronization (ERD) over sensorimotor areas. There was no significant difference between the ERD during the competitive multi-user condition and the single-user condition but considerable inter-individual differences regarding which condition yielded a stronger ERD. Notably, the stronger condition could be predicted from the participants’ MI-induced ERD obtained before the NF blocks. Our findings may contribute to enhance the performance of NF/BCI implementations and highlight the necessity of individualizing context factors.

## 1. Introduction

Advances in health technology such as neurofeedback (NF) or brain–computer interfaces (BCIs) can improve the quality of life for individuals with neurological or psychiatric conditions. While the underlying closed-loop framework may be identical for NF and BCIs, the focus of NF is on improving health conditions, and that of BCIs is on steering technology [1]. Both NF and BCI enable the user to pro-actively self-regulate specific neurophysiological signals. The objective may be to enhance cognitive abilities, support communication, or reduce clinical symptoms. For instance, NF combined with motor imagery (MI) may help to support motor recovery after stroke [2,3,4,5]. 

MI refers to mentally simulated movements that are not succeeded by overt motor output [6]. The neural simulation of action theory [7] states that imagining movements and their actual execution comprise overlapping (sub-)cortical networks in the brain [6,8]. Thereby, the activation of these networks depends on the applied MI strategy, i.e., activation over sensorimotor areas is predominantly elicited by kinesthetic MI (i.e., imagining the feeling of actual executing movements) and less by visual MI (i.e., seeing yourself executing actual movements) [8,9,10]. Since MI is defined by the absence of an overt motor output, it also lacks any sensory feedback on performance. By combining NF and MI, a channel for feeding back information about MI performance can be established. This information can then be utilized by the user to enhance desired activation patterns [11,12,13] and facilitate motor learning or motor recovery [14,15]. 

Many MI-based BCI and NF paradigms involve recording the neurophysiological signal of interest by means of electroencephalography (EEG) [16]. In the context of movement-related paradigms, event-related (de)synchronization (ERD/S), i.e. power changes of rhythmic brain activity captured over sensorimotor areas within the mu (8–12 Hz) and beta (13–30 Hz) frequency range, are prominent neural correlates of MI to control EEG-based NF and BCI paradigms [17,18,19]. EEG is currently the target technique of choice of many BCIs and NFs, because aside from well-established laboratory setups, it can also be recorded using small, low-cost, unobtrusive, and wireless hardware in everyday life [20,21]. This mobile EEG technology allows, for instance, to apply NF and BCI long-term at the user’s home [3,22]. EEG-based BCIs and NFs hold promise for neurorehabilitation and communication, but they are also gaining ground in the gaming industry [23,24]. The latter can be of relevance for clinically oriented applications, as the integration of gaming factors may advance their use. Like any rehabilitative intervention, MI NF practice must be conducted over a long period of time and with many repetitions to be effective [25]. Integrating gaming factors may help to keep up the user’s motivation over the practice period. This is important because motivation plays a central role in the effectiveness of MI NF practice [26]. 

Some EEG studies already touched upon the combination of gaming and MI BCI/NF paradigms by introducing single- and multi-user conditions in collaborative and competitive modes (for a review, see [23]). For instance, Li and colleagues implemented a competitive car racing game in a feasibility study. Although no supporting data were collected at that point, they suggested that a competitive mode would serve as a motivational boost for the user [27]. First comparisons between a single condition and collaborative as well as competitive multi-user conditions were conducted in a study by Bonnet and colleagues [28]. Although they could only find a non-significant trend for higher classification accuracy in their collaborative multi-user compared to their single-user condition, the collaborative multi-user condition was significantly higher rated in terms of enjoyment and motivation by the users. The authors emphasized further that user preference statements regarding social conditions differed across the whole sample, without a consistent pattern emerging. Crucially, user preference statements did not necessarily match the condition with higher classification accuracy, indicating that users’ subjective preferences are not directly reflected in objective performance measures [28]. Nonetheless, the results of this study suggest that social context can possibly affect neural correlates of MI. 

Here we investigated how social context affects MI NF ERD. To this end, a multi-user mobile EEG MI paradigm with an engaging robotic NF was implemented. Participants underwent a single-user condition and a competitive multi-user condition of steering a humanoid robot as far as possible. We tested whether the competitive multi-user condition leads to better performance than the single-user condition. Post hoc, we additionally explored whether some participants performed better in the single-user condition while others performed better in the competitive multi-user condition, with better performance indicated by a stronger relative ERD. We further tested whether specific EEG and behavioral parameters would allow us to predict the social condition in which a participant had the stronger relative ERD. Predictors were ERD in the absence of NF, general performance motivation, and general control beliefs in dealing with technology.

## 2. Materials and Methods

### 2.1. Participants

A total of 27 healthy right-handed young adults participated in this study. All participants reported normal or corrected-to-normal vision, had no history of neurological and psychiatric disorders, and had no prior experience with neurofeedback or MI paradigms. Each participant signed an institutionally approved consent form prior to the experiment. Two data sets were excluded from further analyses, one due to technical issues during the experiment (unstable connection between computer and robot) and one because of non-compliance with task instructions (systematic movements during the MI phase). Consequently, the analysis comprised 25 participants (14 female) aged between 19 and 30 years (*M* ± *SD*: 24.93 years ± 2.95). The study protocol was approved by the Commission for Research Impact Assessment and Ethics of the University of Oldenburg.

### 2.2. Experimental Paradigm

The experiment was conducted in a regular university seminar room. Participants were seated in a comfortable chair with a NAO humanoid robot (SoftBank Robotics, France) in front of them. The participant and robot faced in the same direction. Partitions were used to restrict the participants’ fields of view (cf. Figure 1A). Participants were instructed to take off their shoes to enable the experimenter to monitor potential foot movements during MI. 

All participants performed four experimental blocks. Each block comprised 40 trials. In the first block, foot movements miming walking were physically executed while participants sat on the chair. In the following three blocks, participants remained seated and were instructed to imagine the kinaesthetic sensation of the executed foot movement from a first-person perspective. Between the second and the third block, the channel with the strongest relative ERD was selected for online NF (see Section 2.4 (Data Analysis) for details on online analysis and channel selection). 

Visualizations of the trial timing and NF were based on the Graz MI protocol as implemented in OpenViBE Designer 0.17.1 [29]. Stimuli indicating trial timing were presented on a 2 × 1 meter area in front of the participants using a projector. A double arrow indicated the duration of the task period (5 s). During the 2 s baseline period and the inter-trial interval (4.5 to 6.0 s pseudorandomized in steps of 0.5 s), a fixation cross was presented. Depending on the outcome of the online classification of foot MI ERD, in the third and fourth MI blocks the robot would move. The movement distance was 0, 2.5, 5, or 7.5 cm. Details on feature extraction and classification are given in Section 2.4. The NF was only provided within a maximal time window of 3 s subsequent to the MI interval to avoid interference between feedback and MI. Participants were instructed to try to move the robot as far as possible straight ahead.

There were two different conditions for the MI NF blocks. Either the participant was steering the robot in a single-player mode (single) or participants steered their robot in a race (competitive multi-user) against a pretended opponent (CZ). The order of conditions was counter-balanced across participants. The pretended opponent was only in the room during the block running the competitive multi-user condition. The opponent wore the same EEG equipment as the participant and was barefoot. Other than the pretended opponent, the second robot was in the room in all blocks. The robot of the opponent was computer-controlled and walked each of the possible distances (0, 2.5, 5, 7.5 cm) for one-quarter of the 40 trials in pseudorandomized order. This resulted in a total distance of 150 cm. Ten out of 25 participants won against the pretended opponent in the competitive multi-user condition, that is, their robot covered a greater distance than the opponent’s, while 15 participants lost the race. 

Participants were asked to assess their state motivation on a visual analog scale ranging from “not motivated at all” to “highly motivated” before each experimental block. After each of the three MI blocks, participants rated their MI experience in the previous block on 5-point Likert scales. Ratings were given for easiness of MI (very easy, easy, neutral, difficult, very difficult) and vividness of MI (as vivid as when executing, almost as vivid as when executing, reasonably vivid, moderately vivid, no sensation). The scales also allowed for ratings in between scale intervals. After the main experiment, participants completed questionnaires about performance motivation in general (Leistungsmotivationsinventar Kurzfassung; LMI-K) [30] and general control beliefs in dealing with technology (Kontrollüberzeugungen im Umgang mit Technik; KUT) [31]. The scores of the LMI-K range from 30 to 210 and those of the KUT from 8 to 40. Higher ratings indicate higher performance motivation for the LMI-K and stronger control beliefs for the KUT. Finally, participants were asked if they noticed that their opponent was only pretending to move her robot and were debriefed by one of the experimenters (MD). None of the participants reported noticing that their opponent was only pretending.

### 2.3. Data Acquisition

EEG data were acquired using a wireless, head-mounted EEG system (SMARTING, mBrainTrain, Belgrade, Serbia). The system features a sampling rate of 500 Hz, a resolution of 24 bits, and a bandwidth from DC to 250 Hz. EEG data were collected from 24 scalp sites using sintered Ag/AgCl electrodes with FCz as ground and AFz as reference (see Figure 1C; Easycap, Herrsching, Germany). Electrode impedances were maintained below 10 kΩ before data acquisition. Data acquisition was performed using the OpenViBE acquisition server 0.17.1 [29].

### 2.4. Data Analysis

#### 2.4.1. Online Processing 

EEG data were analyzed online to provide NF. Online analysis consisted of three parts. The first part was performed after the second experimental block (i.e., the first MI block) to determine the channel over sensorimotor areas with the strongest relative ERD using MATLAB (MathWorks, Natick, MA, USA) and the EEGLAB toolbox [32]. The second part consisted of parameter estimation for the subsequent MI NF blocks in OpenViBE [29]. The third part comprised the actual NF delivery during the third and fourth experimental blocks through OpenViBE and the robot. The first part used data from the second experimental block. In this block, participants practiced the MI task without receiving NF while EEG data were collected to train the classifier. After the training block was finished, EEG data were high-pass filtered at 8 Hz (finite impulse response, filter order 826) and subsequently low-pass filtered at 30 Hz (finite impulse response, filter order 220) using EEGLAB toolbox Version 12.0.2.6b for MATLAB (Version: 2011B). This filter range was set to encompass the sensorimotor rhythms mu (8–12 Hz) and beta (13–30 Hz), to which the neural correlate of interest, the event-related desynchronization (ERD), is highly specific [18,19]. Epochs were extracted from -5 to 9 s relative to MI onset, and segments containing artifacts were rejected (EEGLAB function pop_jointprob.m, SD = 3). For each channel the relative ERD was calculated as follows: (1)$$\mathrm{relative}\;\mathrm{ERD}\;\left(t\right)=A\left(t\right)-\frac{R}{R*100}$$
where *R* is the power of a 2 s baseline interval before the task (−3.5 s to −1.5 s relative to the onset of the task), and *A* is the power at time point *t*, with *t* = 0 indicating the onset of MI [18]. For each individual the channel with the strongest relative ERD in the task interval over sensorimotor areas in the training block was selected for online NF (cf. Figure 1C). 

For the second part, the raw data of the selected channel were filtered between 8 and 30 Hz and segmented into baseline intervals ranging from −3.5 s to −1.5 s and MI intervals from 0.5 s to 4.5 s within OpenViBE Version 0.17.1. Segments were spaced to avoid interference of task initialization and expectancy effects with the baseline interval. The resulting intervals were subdivided into 1 s time bins overlapping by 0.9375 s. The logarithmic power of the band-pass-filtered 1 s time windows represented the features for linear discriminant analysis (LDA) classification using sevenfold cross-validation [33]. Based on the classification parameters of the training data set, three border values were calculated corresponding to the lower, middle (median), and upper quartiles of the classification distribution. These borders served as individual thresholds and were used to steer the robot in the subsequent NF blocks.

The third part of the online analysis constituted the actual NF delivery during the two MI NF blocks. Incoming data of the selected channel were 8–30 Hz band-pass filtered in OpenViBE Version 0.17.1. Features were extracted as described above and classified. The single trial classifier output was then compared to the individual thresholds derived from the training data in the second part of the analysis. MI NF classification outputs corresponding to the range below the first quartile of the training classification distribution led to a robot walking distance of 7.5 cm (3 steps), between the first quartile and the second quartile (median) to a walking distance of 5 cm (2 steps), and between the median and the third quartile to a distance of 2.5 cm (1 step), while classifier outputs out of these ranges led to no movement of the robot. 

The online classification accuracy was on average 62% (*SD* = 0.08%) for the training, 63% (*SD* = 0.09%) for the single-user condition, and 64% (*SD* = 0.12%) for the competitive multi-user condition. With 40 trials per condition, online classification accuracies were not significantly above the chance level of 62.5% [34].

#### 2.4.2. Offline Analysis

Offline analysis of the EEG data was conducted using EEGLAB toolbox Version 14.1.1 for MATLAB (Version 9.3; MathWorks, Natick, MA, USA, RRID:SCR_001622). The EEG raw data of all four conditions were high-pass filtered at 1 Hz (finite impulse response, filter order 1650) and subsequently low-pass filtered at 40 Hz (finite impulse response, filter order 166). Filtered data were segmented into consecutive non-overlapping time intervals of 1 s. Segments containing artifacts were rejected (EEGLAB functions pop_jointprob.m, pop_rejkurt.m, both SD = 3). Remaining data were submitted to the extended infomax independent component analysis (ICA) [35] to estimate the unmixing weights of 24 independent components. Components reflecting eye blinks and lateral eye movements were identified using the fully-automated Eye-Catch approach [36]. Components reflecting cardiac activity were identified by visual inspection. Artefactual components were removed from the data, resulting in artifact-corrected EEG data. Data were then low-pass filtered with a finite-impulse response filter and a cut-off frequency of 30 Hz (hamming window, filter order 220, Fs = 500 Hz), and subsequently high-pass filtered with a finite-impulse response filter and a cut-off frequency of 8 Hz (hamming window, filter order 826, Fs = 500 Hz) to encompass the mu and beta frequency bands. Identification of improbable channels was conducted using the EEGLAB extension trimOutlier (https://sccn.ucsd.edu/wiki/EEGLAB_Extensions) with upper and lower boundaries of two standard deviations from the mean across all channels separately for all conditions (mean channels identified: 0.78 ± 0.54, range 0–2). Identified channels were spherically interpolated. Following this, data were segmented from −7 s to 9 s relative to the start of each trial. Artefactual epochs as indicated by the joint probability and kurtosis (EEGLAB functions pop_jointprob.m, pop_rejkurt.m, both SD = 3) within each of the experimental blocks were discarded from further analyses. Additionally, epochs containing values identified as outliers (data points outside 1.5 times the interquartile range above the upper quartile) were removed (0.55 ± 1.18 trials affected, range: 0–5). The relative ERD was calculated for every channel and every MI block (training block, single block, competitive multi-user block) separately following the same procedure as described for online EEG analysis. In contrast to the online analysis, not a single channel but a region of interest (ROI) comprising CZ, CP1, CPz, and CP2 was used for offline analysis (cf. Figure 2C). For statistical analysis, the relative ERD was averaged across a time window covering 0.5 s to 4.5 s with respect to the beginning of the task interval. On average, offline calculations were conducted with 31.57 ± 2.68 trials per subject (range 23–37). 

### 2.5. Statistical Analysis

Prior to statistical analyses, the effects of experimental block order were examined using Fisher’s exact test. To test for the statistical significance of MI NF condition differences in relative ERD, a paired *t*-test between single and competitive multi-user conditions was conducted. 

The finding that a significant difference between the competitive multi-user condition and the single condition was absent in the data was followed up by an exploration of individual differences. To this end, participants were grouped based on their ERD in the MI NF blocks. For the grouping, the relative ERD of the single condition was subtracted from that of the competitive multi-user condition (ΔERD) for each participant. Negative values reflect a stronger relative ERD in the competitive multi-user condition while positive values indicate a stronger relative ERD in the single condition. Negative ΔERD led to an assignment to the group *competitive-gain*, while positive ΔERD led to an assignment to the group *single-gain* (cf. Figure 2B). The final group sizes were 16 subjects for *single-gain* and 9 subjects for *competitive-gain*. To test for differences in relative ERD among MI NF conditions and training conditions, four Bayesian paired sample *t*-tests (reporting Bayes Factors; BF) were conducted in total between each group’s training condition ERD and MI NF ERD (single, competitive multi-user).

Further, to test whether the groups differed in their state motivation, MI intensity, or MI easiness ratings for training, single, and competitive multi-user conditions, three separate (2 × 3) Bayesian mixed ANOVAs with within-subject factor condition (three levels: training, single, competitive multi-user) and between-subject factor group (two levels: *single-gain*, *competitive-gain*) with different questionnaire data as the dependent variable were conducted. In order to evaluate the evidence for the interaction condition × group only, the model containing both main effects and interaction effect was compared to the model with only the main effects (i.e., condition + group/condition + group + condition × group; in JASP version 0.9.2.0, called Inclusion Bayes Factor based on matched models, but also known as Baws Factors; IBF). Moreover, Bayesian independent-sample *t*-tests were performed to characterize both groups regarding their questionnaire data (reporting Bayes Factors; BF).

To identify predictors of the condition with a stronger ERD, a multiple linear regression with forward selection was conducted to predict ΔERD based on relative ERD derived from the training block (ERD_train_), as well as individual scores of the LMI [30] and KUT [31] assessments. 

All numerical values are reported as mean ± SD except where otherwise stated. Effect sizes are reported as Cohen’s d (*d*) for *t*-tests. All Frequentist statistics were calculated as implemented in RStudio (Version 1.1.463; RRID:SCR_001905) [37]. All Bayesian statistics were calculated using the free software JASP using default priors (Version 0.9.2.0; RRID:SCR_015823) [38].

## 3. Results

### 3.1. Comparison of Relative ERD in Single and Competitive Multi-user Conditions

The time-course and topography of MI-induced relative ERD for both MI NF conditions are shown in Figure 2A. To test whether the competitive multi-user condition induced a stronger ERD than the single condition, we performed a paired *t*-test for the ERD of both conditions. We did not find a significant effect (single condition: *M =* −14.85%, *SD* = 16.31%, competitive multi-user condition: *M =* −10.17%, *SD* = 23.3%; *t*_1,24_ = 1.68, p = 0.11, *d* = 0.23). In other words, our data do not provide evidence for the hypothesis that the competitive multi-user condition induces a stronger ERD than the single condition.

### 3.2. Group Comparison of Relative ERD with and without Neurofeedback 

The absence of a significant difference between the competitive multi-user condition and the single condition was followed up by an exploration of interindividual differences in the condition that induced the stronger ERD. To this end, we divided the sample into two groups based on the relative ERD in the single and competitive multi-user conditions, see Figure 2B. Stronger relative ERD in the single condition compared to the competitive multi-user condition led to the assignment to the group *single-gain,* while stronger relative ERD in the competitive multi-user condition compared to the single condition led to an assignment to the group *competitive-gain.* The final group sizes were 16 subjects in *single-gain* and 9 subjects in *competitive-gain*. Prior to further analyses, the effects of experimental block order were examined via Fisher’s exact test. Order had no significant effect on which condition had an ERD gain (OR = 1.53, p = 0.69) or on the outcome (win/lose) of the competitive multi-user condition (OR = 0.32, p = 0.23). Additionally, a Fisher’s exact test showed that gender had no effect on which condition had an ERD gain (OR = 0.97, p > 0.9).

The time-course and topography of MI-induced relative ERD for both groups are shown in Figure 3. Means and standard deviations are given in Table 1. To investigate the differences between training and MI NF conditions, four Bayesian paired sample *t*-tests were conducted. For the *single-gain* group we found very strong evidence of a difference between training and single conditions (BF = 46.0) but only anecdotal evidence for no difference between training and competitive multi-user conditions (BF = 0.46). For the *competitive-gain* group there was moderate evidence for no difference between single and training conditions (BF = 0.33) and weak evidence for no difference between competitive multi-user and training conditions (BF = 0.94). 

The online classification accuracies were, on average, 59% ± 0.08 for the training, 62% ± 0.1 for the single, and 60% ± 0.12 for the competitive multi-user conditions for the *single-gain* group and 65% ± 0.08 for the training, 65% ± 0.07 for the single, and 69% ± 0.09 for the competitive multi-user conditions for the *competitive-gain* group.

To investigate whether the patterns of differences in ERD, that is, stronger ERD for the single condition for group *single-gain* and stronger ERD for the competitive multi-user condition for group *competitive-gain*, are paralleled by differences in state motivation, MI easiness, or MI intensity, we conducted three (2 × 3) Bayesian mixed ANOVAs including the within-subject factor of condition (three levels: training, single, competitive multi-user) and the between-subject factor of group (two levels: *single-gain*, *competitive-gain*) with state motivation, MI easiness, or MI intensity as the dependent variable. In all three cases the comparison of the model containing main effects and the interaction effect with the model containing only the main effects provided moderate evidence against the interaction (motivation: IBF = 0.3, Table 2; MI easiness: IBF = 0.25, Table 3; MI intensity: IBF = 0.23, Table 4; see Section 2.5 (Statistical Analysis) for details). These results indicate that the pattern of differences in relative ERD is not matched by differences in state motivation, MI easiness, or MI intensity.

### 3.3. Group Questionnaire Data

To further characterize the two groups, we compared their general performance motivation (i.e., LMI-K) and general beliefs in dealing with technology (i.e., KUT) by means of two Bayesian independent sample *t*-tests. We found that the groups were comparable regarding their performance motivation (*single-gain: M =* 139.8, *SD* = 32.6, range 36–180; *competitive-gain:*
*M =* 145.78, *SD* = 24.9, range 112-176; BF = 0.41) and their beliefs in dealing with technology (*single-gain:*
*M =* 32.44, *SD* = 7.6, range 13–49; *competitive-gain: M =* 30.11, *SD* = 7.2, range 19–38; BF = 0.46). 

Participants were also asked which social condition they subjectively preferred and how the robotic NF influenced them. In the group *single-gain*, six participants preferred the single condition, six preferred the competitive multi-user condition, and four participants had no preferred condition. In the group *competitive-gain*, two participants preferred the single condition, two preferred the competitive multi-user condition, four participants had no preferred condition, and one participant would have preferred a collaborative condition. Thus, no clear pattern emerged for ERD gain and subjectively preferred social condition. As regards the second aspect, i.e., the influence of the robotic NF, in the group *single-gain,* three participants felt negatively influenced by the robotic NF, six evaluated the influence as neutral, and seven as positive. In the group *competitive-gain* one participant felt negatively influenced by the robotic NF, four evaluated the influence as neutral, and four as positive.

### 3.4. Prediction of Condition with Stronger Relative ERD Based on EEG and Questionnaire Data

In order to pinpoint variables that might be used to prospectively determine the social context condition associated with the strongest relative ERD, we aimed at predicting participants’ ERD group assignments using their KUT and LMI-K scores as well as their relative ERD from the training block, i.e., ERD_train_. We performed a multiple linear regression with forward selection to predict ΔERD, i.e., the difference in relative ERD between single and competitive multi-user conditions, based on LMI-K, KUT scores, and ERD_train_. A significant regression equation was found for ERD_train_ as predictor (F_1,22_ = 8.87, p = 0.007), with an adjusted R^2^ of 0.255. The participants’ predicted ΔERD was equal to 5.91 + 0.41ERD_train_, where ERD_train_ is measured in percent. ΔERD increased 0.41 percent for each percent of ERD_train_. ERD_train_ was the only significant predictor of ΔERD. This result indicates that subjects with a stronger relative ERD in the training block had a stronger relative ERD in the competitive multi-user compared to the single condition (cf. Figure 4) and vice versa.

Scatter plot of ΔERD and ERD_train_ depicted as relative power (%). Participants with a negative ΔERD had a stronger relative ERD in the competitive multi-user condition compared to the single condition (group *competitive-gain* in pink); participants with a positive ΔERD had a stronger relative ERD in the single condition compared to the competitive multi-user condition (group *competitive-gain* in azure). The black line corresponds to the regression line of the significant regression equation with ERD_train_ as the single predictor (see text for details). The grey area indicates the 95% confidence interval. 

## 4. Discussion

In the present study, we aimed to investigate whether social context affects MI NF ERD by comparing single and competitive multi-user conditions. Specifically, we tested whether the competitive multi-user condition leads to better performance than the single-user condition. We did not find a significant difference between the single and competitive multi-user conditions. We further explored the idea that despite the absence of an overall condition effect, participants’ MI NF ERD differs across social context, and that inter-individual differences exist regarding the direction of the difference. Participants were split into two groups based on the MI NF condition with the larger ERD gain. For the *single-gain* group, relative ERD was significantly stronger for the single compared to the training condition, but not stronger for the competitive multi-user compared to the training condition. For the *competitive-gain* group, none of the MI NF conditions showed a stronger relative ERD compared to the training condition. We further tested whether the condition with the stronger relative ERD could be predicted by EEG and/or questionnaire data collected a priori. The only significant predictor emerging from this analysis was EEG, that is, the relative ERD magnitude in the training block. 

### 4.1. Comparison of Relative ERD in Single and Competitive Multi-user Conditions

We did not find a significant difference in relative ERD between single and competitive multi-user conditions in this study. This is in line with results reported by Bonnet and colleagues, who did not find significant differences between single and competitive as well as collaborative multi-user conditions in classification accuracies [28]. To explore whether inter-individual differences may explain the absence of an overall condition effect, we grouped participants based on their MI NF condition with stronger ERD. While the group *single-gain* had a stronger ERD in the single than in the competitive multi-user condition, group *competitive-gain* showed the opposite pattern. The significantly stronger ERD in the single compared to the training condition for the group *single-gain* indicates that for these users, the consideration of social context of an MI NF can boost NF performance. That is, because for participants with lower performance during training, the social context significantly affects their MI ERD, those participants’ MI NF performance is likely to benefit from a single condition compared to a competitive multi-user condition. As already pointed out by Bonnet and colleagues, it is conceivable that these individuals may be distracted by the additional feedback generated by their counterparts or their counterparts in general [28], or the resulting additional cognitive load [39]. One reason for this may be a comparatively low ability to focus attention in these individuals. The impact of potential side effects of social aspects in BCI and NF applications remains to be investigated. Such consideration would be in line with recent developments in the field, i.e., focus towards user-centered [40] and individualized design [41] of MI NF/BCI implementations. 

Notably, the social condition related to a stronger ERD did not consistently match the self-reported preference among participants. This discrepancy between neurophysiological data and self-report is in line with findings by Bonnet and colleagues [28]. They also observed deviations between EEG-based and self-reported user preferences when comparing single, competitive multi-user, and collaborative conditions in a gaming context. Bonnet and colleagues concluded that the relationship of social context and NF performance is influenced by not-yet-identified factors. The present study tested whether general performance motivation and general control beliefs in dealing with technology or state motivation, MI easiness, and MI vividness are likely candidates. Similar to the self-reported condition preference, no group differences were found. These results suggest that the observed differences in relative ERD across conditions and groups are likely not attributable to differences in any of those measures. Thus, the quest for factors explaining ERD and NF performance differences related to social context continues. Assessing general (video) gaming habits and experience could provide interesting insights, as video gaming can affect information-processing abilities potentially relevant for MI NF, including motor skills and visual processing; for a review, see [42].

The focus of this study was on ERD differences between single and competitive multi-user conditions. However, an interesting pattern emerged when additionally considering ERD in the training block, that is, when no NF is given. Here, on a descriptive level, both groups had less pronounced ERD in the training condition than in their stronger social condition, even though the difference was not significant for the group *competitive-gain*. In contrast, the ERD of the weaker social condition was very comparable to the training block ERD for both groups. Future research will show whether this pattern emerged by incidence or whether it is systematic.

### 4.2. Training Block ERD Predicts MI NF Condition with Stronger ERD

Based on the descriptively stronger ERD in the stronger social condition when compared to the training condition and the similarity between the ERD in the training condition and weaker social condition, we further aimed at predicting the MI NF condition associated with a stronger ERD. We included scores indicating general control beliefs in dealing with technology (KUT) and general performance motivation (LMI-K) and based on the relative ERD obtained from the training block (ERD_train_) in the prediction. We found that ERD_train_ predicted the condition with stronger ERD, while test scores made no significant contribution to prediction. Since collecting training data is a prerequisite for most NF setups, this means that a prediction of the social context resulting in a stronger ERD may be possible with only little additional effort. However, whether the regression equation derived from the present data set generalizes to new, online scenarios remains to be tested. 

### 4.3. Absence of Group Differences in Questionnaire Data 

We did not find significant differences between the two groups regarding general performance motivation or general control beliefs in dealing with technology. However, this does not mean that the concepts assessed with these questionnaires play no role for ERD strength in NF paradigms. 

For both measures, it is possible that the degree of abstraction was too high for the performed task. That is, participants may not have felt like they were controlling the robot as a technical device, because they were not directly interacting with the robot, for instance, through a controller, only indirectly through their brain activity. Further, it is possible that participants did not feel challenged by the competitive multi-user condition. Future studies should therefore investigate whether rather artificial situations like competitive NF or NF in general can be related to trait-like characteristics such as general performance motivation, extending, for instance, the work of Zapała and colleagues who found significant associations between MI control and temperamental traits [43].

### 4.4. Implications for MI NF Interventions

In the present study, all results are based on offline analyses. To be able to draw conclusions about online paradigms and their application, the following points need to be considered and potentially adapted.

The experiment was conducted in an unshielded, regular seminar room. Technical equipment in the room and varying ambient noise may have distracted the participants and interfered with the signal recording. During offline analysis, we addressed the potentially increased noise levels in the data likely resulting from the everyday recording with several data preprocessing techniques. During online analysis in future studies, the implementation of appropriate artifact cleaning methods, e.g., [44,45], should be considered. Further, the online channel selection focused on identifying the channel with the strongest relative ERD over sensorimotor areas in the training block. This approach was implemented to ensure a fair comparison between the two MI NF conditions. However, several options might be superior to this approach; an example would be the use of spatial filters in the NF/BCI setup [46,47]. It is also conceivable that an extended training phase, familiarizing participants with the setup and the MI task, could potentially help to improve identification of the channel best reflecting the targeted EEG signature. 

In our setup, the opponent was a colleague pretending to move the robot, which enabled a controlled comparison across participants. It does not, however, allow us to draw conclusions about a setup with two or more real participants. Presumably, factors such as mutual sympathy or demeanor may influence the interplay of participants, as well as their individual performance. In a first step this could, for instance, be investigated with opponents of different gender or different pretended behavior towards the participants, before moving on to two or more real users like in the study of Roc and colleagues, where the influence of different experimenters on participants’ performance was evaluated [48]. 

The online classification accuracies were rather low for this study. This might be due to anatomical specificities of the cortical regions associated with foot MI and a lack of lateralization patterns that could enhance classification accuracies. However, this is difficult to judge as the classification of foot MI relative ERD against baseline EEG activity is rather uncommon in the field. For the present study, it was used to establish a simple and, for the participant, easily accessible link between the NF and the MI task. Much more common and normally associated with higher accuracies is classification based on left vs. right hand MI or hand vs. foot MI (for a review, see [16]). The difficulty of interpreting classification accuracies has been already brought up by Lotte and Jeunet [49] emphasizing that classification accuracies should be handled carefully when quantifying user BCI/NF performance and should be complemented with other metrics. 

Finally, another aspect is that the robotic NF was delivered in a delayed manner to prevent participants from being disturbed or stopping to perform MI before the end of the MI phase. As simulated in a study by Oblak and colleagues, NF performance may be related to an interplay of feedback timing (continuous/intermittent) and self-regulatory mechanisms (cognitive/automatic) [50], prompting the idea that feedback timing is also subject to inter-individual differences or preferences, and also worth encountering in this context.

## 5. Conclusions

Social context constitutes a promising aspect of NF/BCI practice and may help us to enhance NF/BCI performance. Which of the two investigated social contexts generated the stronger ERD differed between individuals; thus, no significant difference between competitive multi-user condition and single condition was found. However, the social context that yielded a stronger ERD could be predicted from the training ERD. Collecting training data is a prerequisite for most NF setups. This result therefore indicates that for the optimization of future MI NF interventions—in particular, those that can operate in single and multi-user mode or, more generally, in individual and group therapy setups—it should be possible to predict the social condition best suited for NF from routinely collected EEG data. 

To conclude, it seems equally promising and challenging to increase MI NF performance on individual levels [51], e.g., by targeting the optimization of signal processing methods [52] or by predicting the user’s MI ability by means of physiological or psychological measures [53,54]. The consideration of context factors for NF/BCI should be a valuable addition in future studies. 

## Figures and Tables

**Figure 1 sensors-20-01620-f001:**
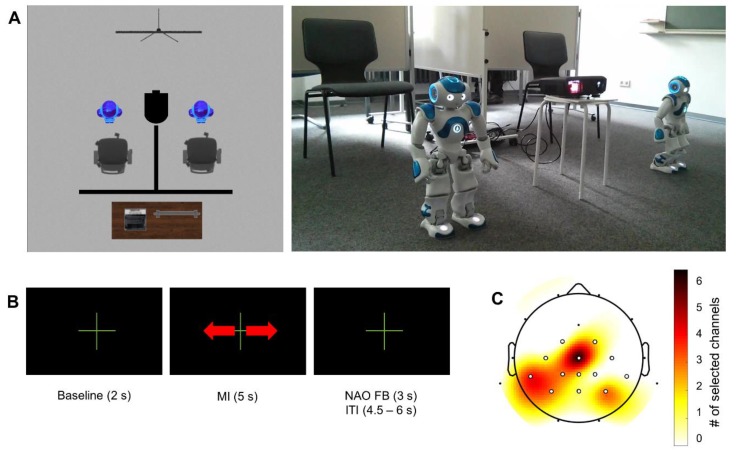
Online scenario. (**A**) Illustration of the experimental setup. The setup consisted of two humanoid robots, two chairs, a projector, and a control station for the experimenters. The chairs were separated from each other and from the control station by partitions such that participants would not see experimenters or their opponent. Robots were placed in front of the chairs. During motor imagery (MI) neurofeedback (NF) blocks (single-user, competitive multi-user), participants were instructed to move their robot as far as possible away from the chair by means of kinesthetic foot MI. (**B**) MI NF trial structure. Each trial started with a fixation cross presented for 2 s (baseline period). Thereafter, a double arrow presented for 5 s indicated the task period. Following the task period, a fixation cross was presented for 4.5 to 6.0 s pseudorandomized in steps of 0.5 s. During the first part of this period (3 s) NAO humanoid robot feedback (NAO FB) could be presented while the second period (4.5 – 6 s) constitutes the inter-trial interval (ITI). (**C**) Distribution of the channel selected online for the robotic MI NF. All 24 electroencephalography (EEG) channels are indicated; channels considered in online channel selection are highlighted in white. For each subject the channel with the strongest relative event-related desynchronization (ERD) in the training block was selected for online NF from this region of interest (ROI). Color-coded is the frequency of channel selection ranging from 0 (channel not selected) to 6 (channel selected as best channel for six participants).

**Figure 2 sensors-20-01620-f002:**
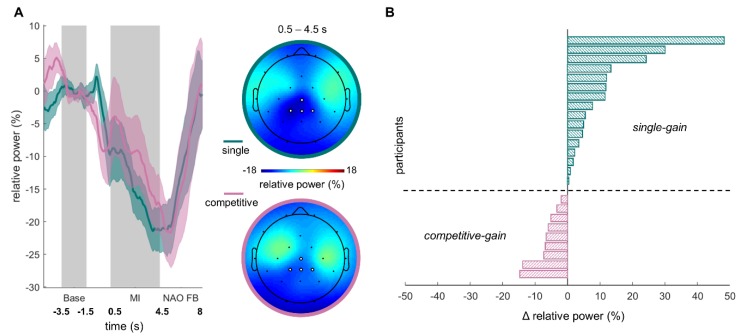
Comparison of relative ERD in single and competitive multi-user conditions. (**A**) Group mean and standard error of relative ERD time courses for MI conditions single (azure) and competitive multi-user (pink). Grey areas indicate the baseline and MI interval. Group mean topographies of relative ERD during the MI interval for both MI NF conditions (top azure frame: single; bottom pink frame: competitive multi-user). The ROI for offline analysis (CZ, CP1, CPz, and CP2) is highlighted in white. (**B**) ΔERD depicted as relative power change (%) (i.e., competitive multi-user minus single condition) to visualize differences between single and competitive multi-user conditions for each participant. Participants with a negative ΔERD had a stronger relative ERD in the competitive multi-user condition compared to the single condition (*competitive-gain* in pink); participants with a positive ΔERD had a stronger relative ERD in the single condition compared to the competitive multi-user condition (*competitive-gain* in azure).

**Figure 3 sensors-20-01620-f003:**
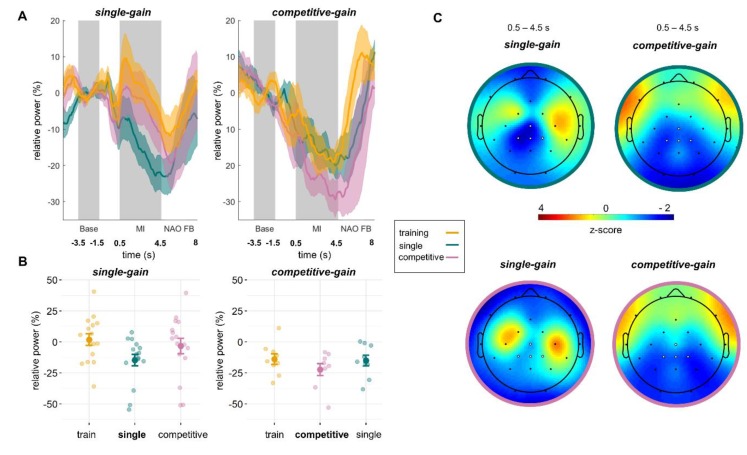
Differences in MI-induced relative ERD between the groups *single-gain* and *competitive-gain*. (**A**) Group mean and standard error of relative ERD time courses for all MI conditions for the *single-gain* (left) and *competitive-gain* (right) groups. Grey areas indicate the baseline and MI interval. (**B**) Group mean, standard error, and single-subject means of relative ERD during the MI interval for all MI conditions for the *single-gain* (left) and *competitive-gain* (right) groups. The NF condition with ERD gain in comparison to the other NF condition is displayed in the middle. (**C**) Group mean topographies of relative ERD during the MI interval for both MI NF conditions (top azure frame: single; bottom pink frame: competitive multi-user) for each group (*single-gain* on the left, *competitive-gain* on the right). The ROI for offline analysis is highlighted in white (CZ, CP1, CPz, and CP2). Topographies were z-transformed within each condition for illustration purposes.

**Figure 4 sensors-20-01620-f004:**
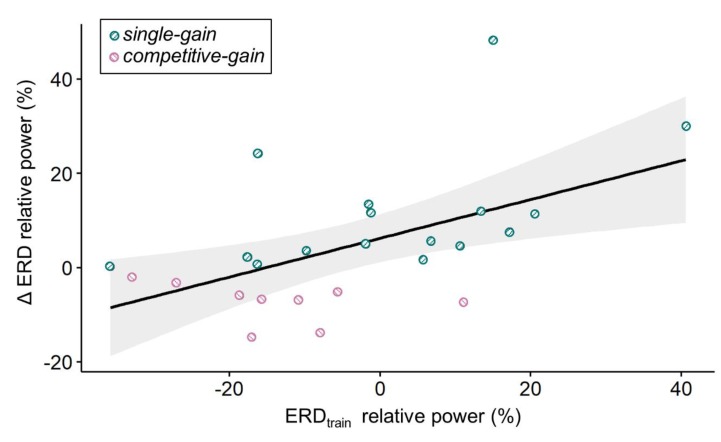
Visualization of the multiple linear regression result. Scatter plot of ΔERD and ERDtrain depicted as relative power (%). Participants with a negative ΔERD had a stronger relative ERD in the competitive multi-user condition compared to the single condition (group competitive-gain in pink); participants with a positive ΔERD had a stronger relative ERD in the single condition compared to the competitive multi-user condition (group competitive-gain in azure). The black line corresponds to the regression line of the significant regression equation with ERDtrain as the single predictor (see text for details). The grey area indicates the 95% confidence interval.

**Table 1 sensors-20-01620-t001:** Descriptive statistics of relative ERD (%) for the conditions training, single, and competitive multi-user for the groups *single-gain* and group *competitive gain.*

	Training		Single		Competitive Multi-User	
	*Single-Gain*	*Competitive-Gain*	*Single-Gain*	*Competitive-Gain*	*Single-Gain*	*Competitive-Gain*
Mean	1.85	−13.88	−14.73	−15.07	−3.32	−22.36
SD	18.50	12.81	18.26	13.16	24.94	14.17
Minimum	−35.85	−33.01	−54.49	−38.12	−50.99	−52.85
Maximum	40.61	11.09	7.74	0.30	39.52	−8.22

**Table 2 sensors-20-01620-t002:** 2 × 3 Bayesian ANOVA for motivation (two levels: *single-gain*, *competitive-gain*; three levels: training, single, and competitive multi-user conditions).

Models	P(M)	P(M|data)	BF _M_	BF _10_	Error %
Null model (incl. subject)	0.20	0.40	2.65	1.00	
Condition	0.20	0.08	0.33	0.19	0.74
Group	0.20	0.42	2.84	1.04	1.53
Condition + Group	0.20	0.08	0.37	0.21	2.56
Condition + Group + Condition × Group	0.20	0.03	0.10	0.06	5.43

*Note.* All models include subject.

**Table 3 sensors-20-01620-t003:** 2 × 3 Bayesian ANOVA for MI easiness (two levels: *single-gain*, *competitive-gain*; three levels: training, single, and competitive multi-user conditions).

Models	P(M)	P(M|data)	BF _M_	BF _10_	Error %
Null model (incl. subject)	0.20	0.05	0.21	1.00	
Condition	0.20	0.59	5.69	12.00	1.26
Group	0.20	0.02	0.09	0.44	0.68
Condition + Group	0.20	0.27	1.50	5.58	1.42
Condition + Group + Condition × Group	0.20	0.07	0.30	1.42	2.53

*Note.* All models include subject.

**Table 4 sensors-20-01620-t004:** 2 × 3 Bayesian ANOVA for MI intensity (two levels: *single-gain*, *competitive-gain*; three levels: training, single, and competitive multi-user conditions).

Models	P(M)	P(M|data)	BF _M_	BF _10_	Error %
Null model (incl. subject)	0.20	0.06	0.23	1.00	
Condition	0.20	0.55	4.86	10.04	0.53
Group	0.20	0.03	0.12	0.52	0.66
Condition + Group	0.20	0.30	1.71	5.49	0.88
Condition + Group + Condition × Group	0.20	0.07	0.29	1.25	2.13

*Note.* All models include subject.

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
