# Peer review of "Challenge Accepted? Individual Performance Gains for Motor Imagery Practice with Humanoid Robotic EEG Neurofeedback"

_sensors, 2020, doi:10.3390/s20061620_

Round 1
Reviewer 1 Report
The manuscript is very well written, well structured and the design of the experiments is scientifically sound.
I have one minor comment. It concerns the section 2.4.1 where it is written that EEG data were high-pass filtered at 8 Hz and low-pass filtered at 30 Hz. It is understandable for everybody who works in the area of EEG processing. However, if a reader is not so familiar with EEG he/she needs some explanation. I would suggest to authors either to add brief explanation or to put a reference to an article or book where the EEG frequency bands are described.
Author Response
I have one minor comment. It concerns the section 2.4.1 where it is written that EEG data were high-pass filtered at 8 Hz and low-pass filtered at 30 Hz. It is understandable for everybody who works in the area of EEG processing. However, if a reader is not so familiar with EEG, he/she needs some explanation. I would suggest to authors either to add brief explanation or to put a reference to an article or book where the EEG frequency bands are described.
We thank the reviewer for this suggestion. We added a descriptive sentence to section 2.4.1 and in addition refer to relevant literature.
After the training block was finished, EEG data were high-pass filtered at 8 Hz (finite impulse response, filter order 826) and subsequently low-pass filtered at 30 Hz (finite impulse response, filter order 220) using EEGLAB toolbox Version 12.0.2.6b for MATLAB (Version: 2011B). This filter range was set to encompass the sensorimotor rhythms mu (8-12 Hz) and beta (13-30 Hz), to which the neural correlate of interest, the event-related desynchronization (ERD), is highly specific to (Cheyne, 2013; Lopes da Silva & Pfurtscheller, 1999). (see lines: 201 - 203)

Reviewer 2 Report
I have to admit that I was quite positively impressed with the study.
I think that the introduction, methods and results sections are very well structured and information is relevant and organized.
It is my opinion that the discussion section is too long and needs to be shortened.
Despite this positive impression of the paper I have several doubts that maybe authors can clarify:
1) Gender: sample has 14 women, why authors did not look into possible gender effect? this is related to my second question.
2) Gaming: everybody knows that gaming experience may have an effect in this type of tasks. Authors do not refer to this aspect as a possible variable affecting performance. Especially women vs men.
3) Authors focused on context and social aspects but what about cognitive processes such as attention? attention may explain some of the results obtained in my opinion and how context may affect focus and attention in general.
These aspects should be mentioned and explained in text maybe with new references support.
Author Response
It is my opinion that the discussion section is too long and needs to be shortened.
We thank the reviewer for this evaluation. We reviewed the discussion critically, but still think that the information provided is important for interpreting the obtained results and for planning subsequent studies. Therefore, we opted to not shorten the discussion.
1) Gender: sample has 14 women, why authors did not look into possible gender effect? this is related to my second question.
We thank the reviewer for this suggestion. A Fisher’s exact test showed that participants gender had no effect on which condition had an ERD gain (OR = 0.97, p > .9). We additionally conducted a point-biserial correlation between ΔERD and participants’ gender and did not find a significant association (t23 = 1.11, p = 0.28). We added the result of the Fisher’s exact test to section 3.2:
Prior to further analyses effects of experimental block order were examined with Fisher’s exact test. Order had no significant effect on which condition had an ERD gain (OR = 1.53, p = .69) or on the outcome (win/lose) of the competitive multi-user condition (OR = 0.32, p = .23). Additionally, a Fisher’s exact test showed that gender had no effect on which condition had an ERD gain (OR = 0.97, p > .9). (see lines: 333 - 335)
2) Gaming: everybody knows that gaming experience may have an effect in this type of tasks. Authors do not refer to this aspect as a possible variable affecting performance. Especially women vs men.
We appreciate the reviewer’s suggestion. Since we did not collect any data about the participants’ gaming habits, we cannot directly address this point. However, we now explicitly mention that information about participants’ gaming habits could be informative in this context, see the discussion section 4.1:
Notably, the social condition related to a stronger ERD did not consistently match the self-reported preference among participants. This discrepancy between neurophysiological data and self-report is in line with findings by Bonnet and colleagues [28]. They also observed deviations between EEG-based and self-reported user preferences when comparing single, competitive multi-user and collaborative conditions in a gaming context. Bonnet and colleagues concluded that the relationship of social context and NF performance is influenced by not yet identified factors. The present study tested whether general performance motivation and general control beliefs in dealing with technology or state motivation, MI easiness and MI vividness are likely candidates. Similar to the self-reported condition preference, no group differences were found. These results suggest that the observed differences in relative ERD across conditions and groups are likely not attributable to differences in any of those measures. Thus, the quest for factors explaining ERD and NF performance differences related to social context continues. Assessing general (video) gaming habits and experience could provide interesting insights, as video gaming can affect information-processing abilities potentially relevant for MI NF including motor skills and visual processing (for a review see, Powers, Brooks, Aldrich, Palladino, & Alfieri, 2013). (see lines: 474 - 476)
3) Authors focused on context and social aspects but what about cognitive processes such as attention? attention may explain some of the results obtained in my opinion and how context may affect focus and attention in general.
We thank the reviewer for raising this interesting point. While our results to not indicate that attention differs in general between the single-user and the multi-user condition, we agree with the reviewer that inter-individual differences in attention may contribute to explaining the differences in ERD gain for both the single- and the competitive multi-user conditions. Indeed, this possibility is already addressed in the discussion, where we explore the possibility that some individuals might be more than others distracted by the feedback, i.e., the robot, of their counterparts. We slightly extended the point made in section 4.1 and now explicitly mentioning attention as a possible contributor.
That is, because for participants with lower performance during training the social context significantly affects their MI ERD, those participants’ MI NF performance is likely to benefit from a single condition compared to a competitive multi-user condition. As already pointed out by Bonnet and colleagues, it is conceivable that these individuals may be distracted by the additional feedback generated by their counterparts or their counterparts in general [28], or the resulting additional cognitive load (Emami & Chau, 2020). One reason for this may be a comparably low ability to focus attention in these individuals. (see lines: 459 - 460)
